# Exploration into Natural Variation Genes Associated with Determinate and Capitulum-like Inflorescence in *Brassica napus*

**DOI:** 10.3390/ijms241612902

**Published:** 2023-08-17

**Authors:** Wei Wan, Haifei Zhao, Kunjiang Yu, Yang Xiang, Wendong Dai, Caifu Du, Entang Tian

**Affiliations:** 1Agricultural College, Guizhou University, Guiyang 550004, China; wwan@gzu.edu.cn (W.W.); zhaohaifeigz@aliyun.com (H.Z.); kjyu1@gzu.edu.cn (K.Y.); 2Guizhou Rapeseed Research Institute, Guizhou Academy of Agricultural Sciences, Guiyang 550006, China; yangxiangyc@zoho.com.cn (Y.X.); daiwendongycs@zoho.com.cn (W.D.); ducaifu_yc@zoho.com.cn (C.D.)

**Keywords:** *Brassica napus*, determinate inflorescence, *BnaAP1*, *BnaTFL1*

## Abstract

*Brassica napus* is a globally important vegetable and oil crop. The research is meaningful for the yield and plant architecture of *B. napus*. In this study, one natural mutant line with determinate and capitulum-like inflorescence was chosen for further study. Genetic analysis indicated that the segregation patterns of inflorescences in the F_2_ populations supported a digenic inheritance model, which was further approved via the BSA-Seq technique. The BSA-Seq method detected two QTL regions on C02 (14.27–18.41 Mb) and C06 (32.98–33.68 Mb) for the genetic control of determinate inflorescences in MT plants. In addition, the expression profile in MT compared with WT was analyzed, and a total of 133 candidate genes for regulating the flower development (75 genes, 56.4%), shoot meristem development (29 genes, 21.8%), and inflorescence meristem development (13 genes, 9.8%) were identified. Then one joint analysis combing BSA-Seq and RNA-Seq identified two candidate genes of *BnaTFL1* and *BnaAP1* for regulating the MT phenotype. Furthermore, the potential utilization of the MT plants was also discussed.

## 1. Introduction

Rapeseed (*Brassica napus* L.) is the world’s third most important source of vegetable oil after palm and soybean [1]. In China, rapeseed accounts for about 85% acreage of oilseed *Brassica*, and is the second most important vegetable oil after soybean [2]. This great importance makes it an ideal model species for theory and application research.

The precise knowledge of the development of inflorescences and flowers is crucial for reproductive success in flowering plants. *Brassica napus* is a simple structure typical of the Brassicaceae and develops an indeterminate raceme inflorescence, individual lateral flowers arising sequentially from an apical inflorescence meristem (IM) [3,4]. The lateral flowers in inflorescence develop according to a well-defined scheme of events that gives rise to a stereotypical floral structure comprising a fixed sequence of concentric whorls with fixed numbers of floral organs (four sepals, four petals, six stamens, and a central pistil) [4]. Once flowers arise on the top of the IM, i.e., the terminal flowers, the growth of inflorescence ceases, which is defined as determinate inflorescence. The situation of the terminal flower has been found in many species, such as in *Arabidopsis thaliana* [5], *Nicotiana tabacum* [6], *Sesamum indicum* [7], *B. juncea* [8], and *B. napus* [9].

The characteristics of inflorescences and flowers are involved in one complex genetic network, in which *LEAFY* (*LFY*), *APETALA1* (*AP1*), and *TERMINAL FLOWER1* (*TFL1*) play key roles. *LFY*, a key transcriptional regulator in the network establishing flower initiation in floral meristem (FM), is activated by genes *AGAMOUS-LIKE 24* (*AGL24*), *SUPPRESSOR OF OVEREXPRESSION OF CO1* (*SOC1*), *MONOPTEROS* (*MP*), and *AINTEGUMENTA* (*ANT*)/*AINTEGUMENTA-LIKE6* (*AIL6*) [10,11,12,13,14]. For the establishment of floral meristem identity, *AP1* plays a key role and is activated by *LFY*, *CAL*, and *LMI2* [15]. The *TFL1* gene has drawn wide interest for its important role in shifting indeterminate inflorescence to determinate inflorescence due to the appearance of a terminal flower at the top of the IM. The terminal flower mutant was first isolated from the mutagenized seeds of *A. thaliana* [5]. Then, the candidate recessive gene of *tfl1* for regulating the terminal flower phenotype in *Arabidopsis thaliana* was mapped on the top arm of chromosome 5 [16] and was cloned [17]. After that, *tfl1*’ in *Nicotiana tabacum* [6] were cloned. In *Sesamum indicum*, the *Sidt1* gene homologous to *Arabidopsis TFL1* was mapped on LG09 [18]. The *Sdt1* homologous to *Arabidopsis TFL1* gene associated with a determinate feature in *Brassica juncea* was mapped to the linkage group B05 [8]. One recent paper on *B. napus* published the discovery of one microspore culture–origin determinate mutation with terminal flowers [9]. The regulator *Bnsdt1*, homologous to Arabidopsis *TFL1*, was fine-mapped on one region of approximately 220 kb, between 16,627 and 16,847 kb on A10, using BC1 and BC3 populations [9]. The cooperative function of these genes could result in differential inflorescence architectures. “A unifying inflorescence model” postulated that *TFL1* could increase vegetativeness (veg) and *LFY* could reduce veg in meristems, leading to different architectures in *Arabidopsis thaliana* [3]. In another model, an increased *TFL1* expression could lead to larger inflorescences with more and longer branches, whereas an increased *AP1* expression could lead to smaller inflorescences with fewer branches and flowers [19].

Except for the traditional techniques for genetic research, such as QTL mapping, many novel techniques based on sequencing have numerously emerged in recent years. RNA-Seq is a recently developed approach to transcriptome profiling using deep-sequencing technologies [20]. The obtained plant transcriptome usually included all coding mRNA and noncoding RNA sequences in the cells of one specific development stage or physiological condition. Plant transcriptome analysis is fast and considerable for providing information on highly expressed genes, differentially expressed genes, new genes for function analysis, and gene screening related to studied traits [21,22,23,24]. Furthermore, the BSA-Seq strategy was one more efficient QTL mapping method compared with the traditional QTL mapping method. Combining the traditional BSA method and the next-generation sequencing (NGS) technique, the BSA-Seq technique has been widely used in QTL mapping for the precise identification of target genes [25,26,27], such as in rapeseed [28], rice [27], maize [29], and so on. Furthermore, the combined analysis of BSA-seq and RNA-seq data enabled the identification of candidate genes [30,31,32].

In the present study, we reported one natural mutant (MT) with determinate and capitulum-like inflorescence in *B. napus*. Then, phenotypic analysis, genetic analysis, RNA-Seq, and BSA-Seq methods were used to identify the candidate genes for regulating the determinate inflorescence. The results would provide a basis for future breeding and gene cloning in *B. napus*.

## 2. Results

### 2.1. Phenotypic Characterization of the Mutant (MT) Plants

One mutant (MT) plant (Figure 1C,D,G,H) was found in the self-pollinated progenies of the wide-type (WT) *B. napus* line of GD605-2 (Figure 1A,B,E,F) in the field. The phenotypes of MT plants and the WT plants were present until the plants started to bolt. Then, at the apex of each inflorescence axis, the indeterminate inflorescence was terminated with a terminal flower in MT plants (Figure 1I,J). The terminal flowers contained one pistil and six stamens, while no petals and seeds would be formed. Each inflorescence axis with buds/flowers at the apex closer to the “terminal flower” was kept clustered together and not prolongated, which finally developed into one capitulum-like head in MT plants (Figure 1C,D,G–J). Each capitulum-like head had about 10.0 ± 0.9 normal flowers/siliques at the top of each inflorescence axis. The MT plants had a plant height of about 158.8 ± 4.3 cm, significantly (*p* = 0.001) lower than the WT plants of 180.9 ± 3.4 cm. Furthermore, no significant difference (*p* = 0.903) in the main inflorescence was detected for the number of siliques between WT plants (58.3 ± 8.2) and MT plants (58.0 ± 9.9).

### 2.2. Genetic Analysis of the MT Phenotype

Genetic analysis was conducted based on the phenotypic characterization of crosses between the MT and WT plants. All the F_1_ plants from four reciprocal crosses between MT and Ningyou7 (29 and 33 plants, respectively), Bakow (29 and 33 plants, respectively), 97,009 (26 and 30 plants, respectively), 97,081 (28 and 15 plants, respectively) displayed a similar phenotype of indeterminate inflorescences to the WT parent plants.

In the F_2_ population from MT × WT, 553 plants with indeterminate inflorescences and 131 plants with determinate inflorescences in 2018 fit into a segregation ratio of 13:3 (χ^2^ = 0.073, *p* = 0.788). In 2020, the F_2_ population from MT × WT was classified into two groups: plants with indeterminate inflorescences (1077 plants) and plants with determinate inflorescences (236 plants), fitting a segregation ratio of 13:3 (χ^2^ = 0.519, *p* = 0.471). Furthermore, another F_2_ population from WT × MT in 2020 also showed the same segregation ratio of 13:3 (843 plants with indeterminate inflorescences versus 172 plants with determinate inflorescences (χ^2^ = 2.169, *p* = 0.141). Therefore, the segregation patterns of inflorescence characterization in the F_2_ populations supported a digenic inheritance model in these crosses.

### 2.3. QTL Mapping by BSA-Seq Technique

The BSA-Seq technique was used to rapidly map the QTL accounting for the determinate inflorescences of MT plants. The “determinate” DNA pool, “indeterminate” DNA pool, and the two parental DNA pools were subjected to Illumina sequencing with an average sequencing depth of 16.25-fold. Then, the clean reads were aligned to the reference genome by using BWA software, and the average proportion of mapped reads to clean reads was 97.08%. After removing low-quality reads, a total of 94.04 Gbp clean data were obtained with an average Q30 rate of 80% and an average onefold coverage ratio of 92.16%. All the obtained clean data were used to develop SNP and InDel. A total of 1,500,703 SNPs and 945,276 SNPs were identified between the two DNA pools, respectively. Furthermore, a total of 400,329 InDel between two parents and 280,916 InDel between two DNA pools were identified. The ΔSNP-index and ΔInDel-index were used for QTL mapping of the candidate intervals for determinate inflorescence. As a result, one overlapped region was identified on C02, which had a size of 4.18 Mb (14.27–18.45 Mb) by ΔSNP-index and 4.18 Mb (14.23–18.41 Mb) by ΔInDel-index (Figure 2), respectively. On C06, two other overlapped regions were identified on C06, which had sizes of 0.7 Mb (32.98–33.68 Mb) by ΔSNP-index and 0.72 Mb (32.97–33.69 Mb) by ΔInDel-index, respectively. Finally, these two overlapped regions on C02 (14.27–18.41 Mb, 320 candidate genes) and C06 (32.98–33.68 Mb, 117 candidate genes) were chosen as the candidate regions for the genetic control of determinate inflorescences in MT plants.

### 2.4. Gene Expression Profile Analysis

A total of six cDNA libraries were constructed and sequenced, where an average of 48,705,785 and 46,319,327 raw reads were generated from the WT and MT libraries, respectively (Table 1 and Appendix A). After removing low-quality reads, adapter polluted reads, and higher-N-content (>5%) reads, an average of 47,184,448 (WT) and 44,944,151 (MT) clean reads were obtained. After blasting the reference genome of *B. napus* and filtering the genes that only contained one exon or encoded short peptide chains (<50 amino acid residues), a total of 86,026 genes were revealed through blasting the reference genome using DESeq2 (v1.6.3). It was found that 6309 genes were differentially expressed, among which, 4103 were downregulated and 2209 were upregulated in MT compared with WT (Appendix A). Based on the FPKM values, the upregulated and downregulated genes between WT and MT were also revealed via a hierarchical clustering analysis in Appendix A. To verify the expression of DEGs detected by RNA-Seq, 16 candidate genes (DEGs) regulating the MT phenotype were randomly chosen for validation by qRT-PCR. The data obtained via qRT-PCR was consistent with the RNA-Seq results (Appendix A), suggesting the reliability of the transcriptome database.

To functionally annotate the *B. napus* transcriptome, the 6309 DEGs were blasted in search of Gene Ontology (GO) and Kyoto Encyclopedia of Genes and Genomes (KEGG). Finally, 4548 DEGs were successfully annotated, among which 4522 (71.68%) were in GO and 2127 (33.71%) were in KEGG (Appendix A). For the GO classification analysis of DEGs, 4522 genes were assigned into three main GO functional categories (cellular component, biological process, and molecular function) and then divided into 55 sub-categories (Appendix A). For the KEGG analysis of gene functions, all the annotated 6309 DEGs were assigned to 128 pathways (Appendix A) based on the KEGG database. Several DEGs were assigned to more than one sub-category.

### 2.5. Candidate Genes for Regulating the Determinate Inflorescence

Following the annotation of GO and KEGG, the potential DEGs for MT phenotype were further blasted with the database of *Arabidopsis thaliana* (https://www.arabidopsis.org/index.jsp) on 20 June 2023 and NCBI (https://www.ncbi.nlm.nih.gov/) on 23 June 2023 for detailed information. A total of 133 candidate genes for regulating the flower development (75 genes, 58.6%), shoot meristem development (29 genes, 22.7%), and inflorescence meristem development were identified (13 genes, 9.8%) (Appendix A). Several genes were involved in more than one sub-category.

To integrate the results of BSA-Seq and RNA-sequencing, we perform an alignment analysis between the 133 candidate genes potentially related to the determinate inflorescence of the MT plants and the reference genome of *B. napus* (https://www.genoscope.cns.fr/brassicanapus/) on 27 June 2023 using the BLAST-like alignment tool [33]. Eight DEGs corresponding to seven genes (Table 2) located on C02 between 14.27 Mb and 18.41 Mb were detected, among which, the widely recognized genes for regulating determinate inflorescence of BnaC02g02900D encoding *TERMINAL FLOWER 1* (*TFL1*) (ChrC02: 1,320,657–1,321,719) were included [9,34]. In addition, three DEGs on C06 (Table 2) between 32.98 Mb and 33.68 Mb were detected, among which, only one DEG of BnaC06g25500D encoding *APETALA1* (*AP1*) (C06: 27,150,336-27,153,999) was related to the inflorescence development. Then, the two candidate genes were renamed as *BnaTFL1* and *BnaAP1*. The expression level of *BnaTFL1* and *BnaAP1* significantly increased from the 5-leaf stage and started to decrease from the 7-leaf stage in WT plants (Figure 3). In addition, the expression level of *BnaTFL1* and *BnaAP1* was significantly decreased from the 5-leaf stage to the 7-leaf stage in MT plants compared with WT plants (Figure 3).

## 3. Discussion

In the present study, one natural mutation with determinate inflorescences characterizing terminal flower and capitulum-like inflorescences in Brassiceae were reported. The genetic mechanism of determinate phenotype has been studied for many years. In some diploid species, such as *Arabidopsis thaliana* [5] and *Sesamum indicum* [7], *TERMINAL FLOWER 1* (*TFL1*) or its homologs was reported to solely regulate the phenotype of determinate inflorescence. While in some tetraploid, the digenic inheritance model was more popular. In *Nicotiana tabacum* [6], two gene copies of *CEN*/*TFL1* and *NCH* were cloned and verified using the Southern blot hybridization method, while no further research about the function of these two genes and their interaction in regulating the mutant phenotype was found. In *Brassica napus* [9], *Bnsdt1*, one homolog of the *TFL1* gene, was obtained, and the gene cloning and functional analysis of another gene of *Bnsdt1* is currently still undergoing study.

In our study, the segregation patterns of inflorescences in the F_2_ populations supported a digenic inheritance model, which was further approved by the BSA-Seq technique. *TERMINAL FLOWER 1* (*TFL1*) has been proven to be responsible for controlling the determinate inflorescences in many species [5,6,7,8,9,35], while *BnaTFL1* was mapped on C02 in the present study, which might be different from these genes on B05 of *B. juncea* [8] and on A10 of *B. napus* [36] in Brassiceae. In addition, *AP1* had previously been identified as the gene controlling the development of inflorescence meristem, and the floral meristem [19], *BnaAP1*, on C06 was first reported as the candidate gene for controlling determinate inflorescences. Quite different from the reported phenotype of determinate inflorescences, the capitulum-like inflorescences were also found in GD605-2. The formation of capitulum-like inflorescences had once been well exploited by two models based on *TFL1* and *LFY* [3] or *TFL1* and *AP1* [19], in which the specific combination of gene expression levels resulted in the emergence of different capitulum-like inflorescences.

The mutant with determinate and capitulum-like inflorescence could provide unique materials for the basic research of the development of inflorescence in *B. napus*. The results would provide a basis for future breeding and gene cloning in *B. napus*.

## 4. Materials and Methods

### 4.1. Plant Materials and Field Experiment

In March 2014, one mutant (MT) plant with determinate and capitulum-like inflorescence was found. The MT plant was found in the self-pollinated progenies of one breeding line GD605-2 (F_7_) (wild-type, WT). In October 2017, twenty plants for each of the MT (F_4_) and WT (F_7_) lines were planted in the field of Guizhou University, Guiyang, China.

### 4.2. Investigation and Analysis of the Agronomic Traits of MT and WT Plants

In April 2019, twenty plants for each of the MT and WT plants were used for the investigation of the number of flowers/siliques in the capitulum-like head, plant height, and number of siliques. The number of flowers/siliques, plant height, and number of siliques were identified using mean ± standard deviation. The comparisons of plant height and number of siliques between the MT and WT plants were performed using Mann–Whitney test. The genetic analysis of F_2_ plants was conducted using the chi-square test method.

### 4.3. BSA-Seq Analysis

The BSA-Seq method was used for mapping the genes of regulating the phenotype of determinate inflorescence. The “determinate” bulk was made by mixing an equal amount of DNA from 10 F_2_ plants with determinate inflorescence, while the “indeterminate” bulk was formed from 10 F_2_ plants with indeterminate inflorescence. The two bulks and the DNA samples of the parental lines of WT plants and MT plants were sequenced on an Illumina HiSeq™2000 platform (Beijing Biomarker Biotechnology Co., Beijing, China). The low-quality reads containing adaptors were filtered. The reads with more than 10% of missing bases and more than 50% of bases with a Q-score lower than 10 were filtered, and the clean reads thus obtained were mapped to the *Brassica napus* reference genome (Genoscope v4.1, http://www.genoscope.cns.fr/brassicanapus/data/) on 18 May 2023 using BWA (version 0.5) [27,37]. SNP-calling and SNP annotation were performed using the professional software of ANNOVAR (version Rev.502) [37,38], MarkDuplicatePicards (http://sourceforge.net/projects/picard/), GATK (version 4.1.9) [39], and SnpEff (version 1.9.6) [40]. The differences in allele frequency between bulked pools were used to calculate the SNP index for identifying the candidate regions of the genome associated with determinate inflorescences [27,28,41]. In detail, Δ(SNP index) was calculated by sliding window analysis among the genome within 1 Mb-wide windows and 1 kb at each step [26]. In total, all those analyses were performed with the related tools on the online open platform of BMKCloud (http://www.biocloud.com/) on 5 July 2023.

### 4.4. RNA Extraction, Preparation, Sequencing, and Data Analysis

Total RNA (2 µg) was extracted from the shoots (0.2–0.5 g) of three independent plants for each of the WT and MT lines in the 5-leaf stage using the TRIzol kit (Invitrogen, Carlsbad, CA, USA), according to the manufacturer’s instructions. The RNA purity was checked using the Kaiao K5500^®^Spectrophotometer (Kaiao, Beijing, China), and the RNA integrity and concentration were assessed using the RNA Nano 6000 Assay Kit for the Bioanalyzer 2100 system (Agilent Technologies, Santa Clara, CA, USA). Then, the six RNA samples were sent to the ANOROAD GENOME company (Beijing, China) for the construction of cDNA libraries and Illumina deep sequencing according to the paper by Wang et al. [42]. The raw RNA-sequencing data were filtered by a Perl script, following the steps by Wu et al. [43] and from our earlier study [32].

### 4.5. Identification and Annotation of Differentially Expressed Genes (DEGs)

DESeq2 v1.6.3 was used for differential gene expression analysis between MT and WT with three biological replicates under the theoretical basis, obeying the hypothesis of the negative binomial distribution for the value of count. The *p*-value was corrected by the BH method. Genes with *p* ≤ 0.05 and |log2_ratio| ≥ 1 were identified as differentially expressed genes (DEGs) [44]. The DEGs obtained were further annotated with Gene Ontology (GO, http://geneontology.org/) on 11 May 2023 and analyzed by KEGG (Kyoto Encyclopedia of Genes and Genomes, http://www.kegg.jp/) on 22 May 2023 [45,46]. The GO enrichment of DEGs was implemented by the hypergeometric test, in which the *p*-value is calculated and adjusted to produce the q-value, and the data background is the genes in the whole genome. GO terms with q < 0.05 were considered to be significantly enriched. GO enrichment analysis was used to determine the biological functions of the DEGs. The KEGG enrichment of the DEGs was determined by the hypergeometric test, in which *p*-value was adjusted by multiple comparisons to produce the *p*-value. KEGG terms with q < 0.05 were considered to be significantly enriched.

### 4.6. Quantitative Real-Time PCR (qRT-PCR) Analysis

Quantitative real-time PCR (qRT-PCR) was used to verify the transcript levels of the RNA-Seq results. Total RNA was extracted using the TRIzol kit (Invitrogen, Waltham, MA, USA), according to the manufacturer’s instructions. Then, the cDNA was synthesized by reverse transcription using PrimeScript RT reagent kits with gDNA Eraser (Takara, Dalian, China) according to the manufacturer’s instructions. Sixteen gene-specific primers for qRT-PCR were designed based on reference unigene sequences randomly chosen from the DEGs using Primer Premier 5.0. Real-time PCR was conducted using SsoAdvancedTM Universal SYBR Green Supermix (Hercules, CA, USA) according to our earlier research [32]. The 2-ΔΔCt algorithm was used to calculate the relative level of gene expression. The β-actin gene was used as the internal control, and the WT samples served as the control. All qRT-PCR were performed with three biological replicates, and run on a Bio-Rad CFX96 Real-Time System (Bio-Rad, Hercules, CA, USA).

## 5. Conclusions

In the present study, we reported one natural mutation with determinate and capitulum-like inflorescence in *B. napus*. Through genetic analysis and BSA-Seq analysis, two QTL regions on C02 and C06 were detected. To better dig the candidate genes within the QTL regions, RNA-Seq analysis was conducted. Finally, one joint analysis combing BSA-Seq and RNA-Seq identified two candidate genes of *BnaTFL1* and *BnaAP1* for regulating the MT phenotype.

## Figures and Tables

**Figure 1 ijms-24-12902-f001:**
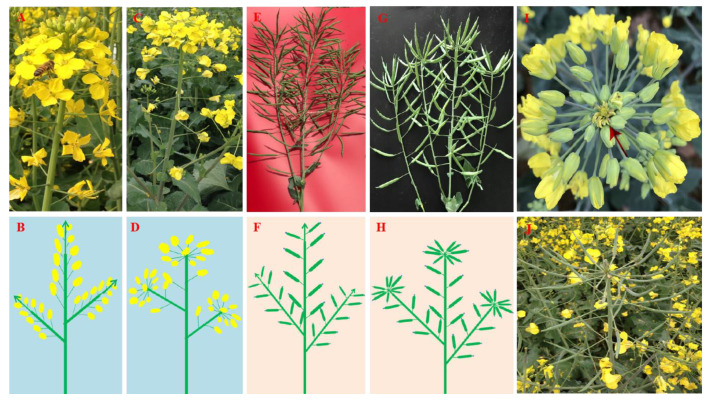
The phenotype analysis between mutant (MT) plant and wild-type (WT) plant in flowering and pod development period. (**A**) One normal indeterminate inflorescence; (**B**) diagram of (**A**); (**C**) one mutant branch with determinate inflorescence; (**D**) diagram of (**C**); (**E**) normal branches with siliques; (**F**) diagram of (**E**); (**G**) mutant branches with siliques; (**H**) diagram of (**G**); (**I**) the top of mutant inflorescence with a terminal flower (red arrow); (**J**) the top of mutant plants in silique developing stage. (**B**,**D**,**F**,**H**) are not diagrams but schemes (diagrams of flowers/inflorescences represent an apical view, and arrows in the diagram picture represent indeterminate growth).

**Figure 2 ijms-24-12902-f002:**
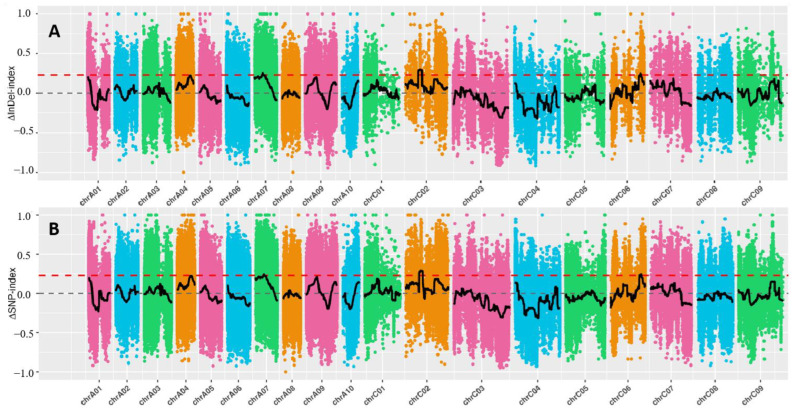
QTL mapping results were obtained via ΔSNP-index (**A**) and ΔSNP-inDel (**B**) methods of the BSA-Seq technique.

**Figure 3 ijms-24-12902-f003:**
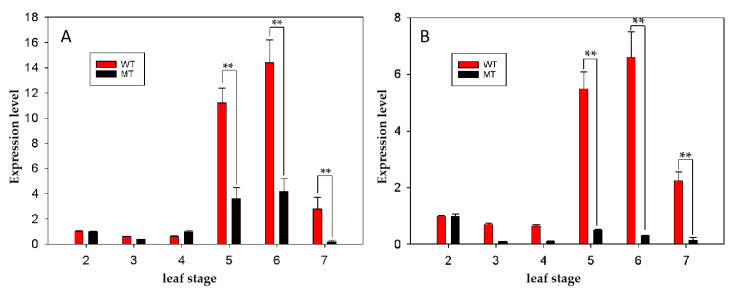
The expression level of *BnaTFL1* (**A**) and *BnaAP1* (**B**) in the shoots of MT and WT lines from the 2-leaf stage to the 7-leaf stage. The number from 2 to 7 on the X-axis indicates the sampling stages from the 2-leaf stage to the 7-leaf stage. ** indicates the expression level is significantly different at the level of *p* < 0.001 using *t*-test.

**Table 1 ijms-24-12902-t001:** Summary of transcriptome sequencing data.

Items	WT (Mean)	MT (Mean)
Raw Reads Number	48,705,785	46,319,327
Raw Bases Number	7,305,867,800	6,947,899,100
Clean Reads Number (%)	47,184,448 (96.88)	44,944,151 (96.96)
Clean Bases Number	7,077,667,200	6,741,622,600
Low-Quality Reads Number (%)	344,567 (0.71)	360,269 (0.78)
Mapped Reads (%)	41,135,862 (87.18)	39,824,899 (88.60)
Unmapped Reads	6,048,586	5,119,252
Multi-Map Reads (%)	6,988,386 (14.81)	6,857,621 (15.26)
Ns Reads Number (%)	3115 (0.01)	3694 (0.01)
Adapter Polluted Reads Number (%)	1,173,656 (2.41)	1,011,214 (2.18)
Raw Q30 Bases Rate (%)	93.84	93.67
Clean Q30 Bases Rate (%)	94.14	94.01
Exon (%)	15,111,070 (94.36)	14,619,066 (94.53)
Intron (%)	249,023 (1.56)	221,680 (1.43)
Intergenic (%)	649,815 (4.07)	623,039 (4.04)
Novel Transcripts	160,798	129,446

**Table 2 ijms-24-12902-t002:** The DEGs are located in the QTL regions on C02 and C06.

Gene ID in Darmor-Bzh	Arabidopsis ID	E-Value	Gene Name	Gene Function
BnaC02g02980D	AT5G03680	0	*PETAL LOSS*	Organ initiation and orientation
BnaC02g02940D	AT2G18960	4 × 10^102^	*OPEN STOMATA 2*	Regulation of stomatal movement
BnaC02g02910D	AT2G46030	3 × 10^85^	*UBIQUITIN-CONJUGATING ENZYME 6*	Ubiquitin-dependent protein catabolic process
BnaC02g02900D	AT5G03840	0	*TERMINAL FLOWER 1*	Controls inflorescence meristem identity
BnaC02g02880D	AT5G64140	1 × 10^55^	*RIBOSOMAL PROTEIN S28*	Ribosomal small subunit assembly and translation
BnaC02g02870D	AT5G03860	0	*MALATE SYNTHASE*	Encodes a protein with malate synthase activity
BnaC02g02830D	AT5G03940	0	*54 CHLOROPLAST PROTEIN*	Protein import into chloroplast thylakoid membrane
BnaC02g02820D				
BnaC06g25530D	AT1G68990	0	*MALE GAMETOPHYTE DEFECTIVE 3*	Transcription of mitochondrial genes
BnaC06g25500D	AT1G69120	0	*APETALA1, AP1*	Inflorescence meristem identity; specifies floral meristem and sepal identity
BnaC06g25460D	AT3G08900	2 × 10^48^	*REVERSIBLY GLYCOSYLATED POLYPEPTIDE 3*	UDP-L-arabinose metabolic process

## Data Availability

The data produced in this study are available in the article and Appendix A. The raw Illumina sequence data generated for this study were deposited in the NCBI Sequence Read Archive (SRA), under BioProject PRJNA592310.

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
