# Peer review of "Exploration into Natural Variation Genes Associated with Determinate and Capitulum-like Inflorescence in Brassica napus"

_ijms, 2023, doi:10.3390/ijms241612902_

Round 1
Reviewer 1 Report
The reviewed paper is devoted to the characterization of an abnormal phenotype in Brassica napus. It is concise and reports some results which, however, represent some intermediate stage of yet unfinished research.
The authors applied adequate methods and their work seems quite logical. However, almost every part of it contains some major flaws which require a serious elaboration.
1. The language of this text needs editing. I would recommend authors to have their manuscript checked by either a native speaker or some specialized language service prior to resubmission.
2. The morphogical description of mutant vs. wild-type phenotype deserves improvement. Some statements are not clear. The authors claim that the mutant plants produce a terminal flower, which is a very unusual feature. However, neither photo images nor schemes do not illustrate the presence of this ectopic flower. It might be a great improvement if authors could illustrate the morphology of this flower itself by photo image(s) and/or diagram(s).
3. All anatomical part seems irrelevant and, in my opinion, should be omitted. Most of sections in Figure 2 do not traverse through the apex itself, so the apex is not illustrated. What is stated to be the initiating terminal flower in Fig. 2e-f by authors is simply an apical meristem, probably already initiating one or two lateral floral primordia. The stages which authors chose for sampling are too early to capture the emergence and development of a terminal flower. However, this part is not that necessary for the understanding of this paper. I would say that images of a terminal flower are more desired that anatomical sections.
4. The interpretation of results of crosses and hybrid analysis should be elaborated as well. As there are probably two genes involved, the mutant phenotype should not be referred to as dominant or recessive.
5. I cannot understand why authors did not compare sequences of putative causative genes between MT and WT plants. It might be informative enough.
It should be noted that differences in expression level do not prove that certain genes are indeed involved in the inflorescence determinacy. To prove it unequivocally, there should be a special research of how these candidate genes influence the phenotype of transgenic plant like Arabidopsis. The rest is only a hypothesis which remains still somewhat unfounded although highly likely.
6. The results of the statistical procedures need to be included and discussed in more details. At the moment, it is even unclear which tests authors applied.
More comments and corrections are available in the manuscript file (see attached).
I would recommend authors to seriously revise their paper. After this it can be reviewed again to decide whether it is acceptable for publication in the journal. I also wish authors good luck.

The language of this paper needs a deep revision. At the moment, some of pieces are difficult to understand.
Author Response
Thanks a lot for your excellent suggestions, and we had revised the Ms as follows:
- The language of this text needs editing. I would recommend authors to have their manuscript checked by either a native speaker or some specialized language service prior to resubmission.
Reply: Thanks a lot for your suggestions. We fully accepted your comments and has edited the language with the revision mode.
- The morphogical description of mutant vs. wild-type phenotype deserves improvement. Some statements are not clear. The authors claim that the mutant plants produce a terminal flower, which is a very unusual feature. However, neither photo images nor schemes do not illustrate the presence of this ectopic flower. It might be a great improvement if authors could illustrate the morphology of this flower itself by photo image(s) and/or diagram(s).
Reply: Thanks a lot for your good suggestions. We fully accepted your comments and modified Figure 1. on the original MS.
Figure 1 in the revised version–Figures 1I and J were added to better illustrate the presence of a terminal flower.
- All anatomical part seems irrelevant and, in my opinion, should be omitted. Most of sections in Figure 2 do not traverse through the apex itself, so the apex is not illustrated. What is stated to be the initiating terminal flower in Fig. 2e-f by authors is simply an apical meristem, probably already initiating one or two lateral floral primordia. The stages which authors chose for sampling are too early to capture the emergence and development of a terminal flower. However, this part is not that necessary for the understanding of this paper. I would say that images of a terminal flower are more desired that anatomical sections.
Reply: Thanks a lot for your good suggestions. We fully accepted your comments and Figure 2 was deleted from the original MS.
- The interpretation of results of crosses and hybrid analysis should be elaborated as well. As there are probably two genes involved, the mutant phenotype should not be referred to as dominant or recessive.
Reply: Thanks a lot for your good suggestions. We fully accepted your comments and the description on the original MS referred to as dominant or recessive in F1 was deleted.
- I cannot understand why authors did not compare sequences of putative causative genes between MT and WT plants. It might be informative enough.
It should be noted that differences in expression level do not prove that certain genes are indeed involved in the inflorescence determinacy. To prove it unequivocally, there should be a special research of how these candidate genes influence the phenotype of transgenic plant like Arabidopsis. The rest is only a hypothesis which remains still somewhat unfounded although highly likely.
Reply: Thanks a lot for your good suggestions. While the work related to gene cloning, sequence analysis and functional verification of the two candidate genes is going on by another student. Based on our overall considerations, we would publish these results in next paper one year later. Thanks, how do you think about that?
- The results of the statistical procedures need to be included and discussed in more details. At the moment, it is even unclear which tests authors applied.
Reply: Thanks a lot for your good suggestions. We fully accepted your comments and revised them on the original MS.
In the section of “4.2. Investigation and analysis of the agronomic traits of MT and WT plants” of “4. Materials and methods”–the statistical method was added.
In the first paragraph of “ 3. Discussion”–more discussion related to the statistical result is discussed.
Other response to the first reviewer for the suggestions in Ms:
Firstly, we are very grateful for your detained revisions of these language, and we had completely changed according to your suggestions.
- For the gene name, “c”means c genome in our original Ms, while we change BncAP1 and BncTFL1 into BnaAP1 and BnaTFL1 in the whole Ms for that Bn couldn’t distinguish Brassica napus and Brassica nigra, after seeking advice from some experts.
- For the p value in “In the F2 population from MT×WT, 553 plants with indeterminate inflorescences and 131 plants with determinate inflorescences in 2018, fitting a segregation ratio of 13:3 (c2=0.073, p= 0.788).“
Yes, my student recalculate the p value and it is 0.788 by Chi square test.

Reviewer 2 Report
The presented article is devoted to the description and multilateral analyses of new rapeseed mutation characterized by determinate and capitulum like inflorescence. Now an analysis of morphological character is a rare topic of investigations. And it is a great pity, because now we have a lot of new methods, which help to reveal mechanisms of traits formation, as it was done in the presented investigation. This new data, in its turn, helps to understand the mechanisms of plants’ development.
In the current investigation rapeseed mutant with determinant inflorescence was analyzed by microscopy and it was discovered that the inflorescence apex began to split and developed into one terminal flower, starting from five-leaf stage. Hybrid analyses revealed digenic inheritance of the character. The BSA-Seq method also detected two QTL regions associated with this phenotype. BSA-Seq and RNA-Seq joint analysis identified two candidate genes: BncTFL1 and BncAP1 for described phenotype. So, multilateral analyses of the mutant were completed.
The article can be published after some English language edditing.
In some sentences words are incorrectly matched
Author Response
Thanks a lot for your excellent suggestions, and we had revised the Ms as follows:
The article can be published after some English language edditing.
Reply: Thanks a lot for your suggestions. We fully accepted your comments and has edited the language with the revision mode.

Reviewer 3 Report
The study of Wan and colleagues is interesting. The main aim of the study and the adopted methodology are well described. Moreover it has generated a significant amount of data which will be very valuable to those working in this area. On the other hand, I must report several inaccuracies that in my opinion render this manuscript unsuitable for publication on IJMS as it stands. Above all, in my opinion the manuscript completely miss the paragraph of discussion. This paragraph seems to be an extension of the conclusions, it's really poor too! Moreover, I do not find any reference to the public repository archive where raw sequencing data were deposited (for both RNA and DNA sequencing), e.g. NCBI SRA or similar. The authors are required to include a Data Availability Statement in their article. In addition in subsections 2.2 and 2.3 I found some refuses: on row 143 “crosss”; on row 153 “945,276 is repeated twice; on row 159 “wihch”
Author Response
Thanks a lot for your excellent suggestions, and we had revised the Ms as follows:
The study of Wan and colleagues is interesting. The main aim of the study and the adopted methodology are well described. Moreover it has generated a significant amount of data which will be very valuable to those working in this area. On the other hand, I must report several inaccuracies that in my opinion render this manuscript unsuitable for publication on IJMS as it stands. Above all, in my opinion the manuscript completely miss the paragraph of discussion. This paragraph seems to be an extension of the conclusions, it's really poor too!
Reply: Thanks a lot for your suggestions. We fully accepted your comments and has edited the Ms as follows:
In the first paragraph of “ 3. Discussion”–more discussion related to the statistical result is discussed.
Moreover, I do not find any reference to the public repository archive where raw sequencing data were deposited (for both RNA and DNA sequencing), e.g. NCBI SRA or similar. The authors are required to include a Data Availability Statement in their article.
Reply: Thanks a lot for your suggestions. We fully accepted your comments and has edited the Ms as follows:
In the Lines 318-3200 of the revised version–“Data availability statement” has been added.
In addition in subsections 2.2 and 2.3 I found some refuses: on row 143 “crosss”; on row 153 “945,276 is repeated twice; on row 159 “wihch”
Reply: Thanks a lot for your suggestions. We fully accepted your comments and has edited the Ms as follows:
In the Lines 124, 134, 140 of the revised version–the refuses has been corrected.

Round 2
Reviewer 1 Report
As I may see, the authors have significantly revised their paper.
I am glad that the authors decided to remove the part devoted to anatomical study of plant apices. It might be also a good idea to elaborate molecular part of this work and present it in a further publication to make it more founded.
I have several minor comments and suggestions how to improve this work more.
1. In Fig. 1I, please add arrow/arrowhead indicating a terminal flower. Please also remove word 'one', as a terminal flower is by default unique. Please also note that Fig. 1B, D, F and H are not diagrams but schemes (diagrams of flowers/inflorescences represent an apical view).
2. Fig. 3: Please indicate which statistical test was applied for pairwise comparison in a caption.
3. Line 237: was reported --> were reported
There are some minor language flaws which will most probably be corrected during the preparation of a manuscript for publication.
In its updated form, this paper can be accepted for publication in IJMS.
Some minor corrections are inevitably required but can be introduced in the course of paper preparation for publication by the Editorial office team.
Reviewer 3 Report
The authors accepted my previous suggestions